# Strain Characterization in Two-Dimensional Crystals

**DOI:** 10.3390/ma14164460

**Published:** 2021-08-09

**Authors:** Shizhe Feng, Zhiping Xu

**Affiliations:** Applied Mechanics Laboratory, Center for Nano and Micro Mechanics, Department of Engineering Mechanics, Tsinghua University, Beijing 100084, China; burst_fsz@163.com

**Keywords:** 2D crystals, strain field, geometrical phase analysis, bond distortion, virial stress, atomistic simulations

## Abstract

Two-dimensional (2D) crystals provides a material platform to explore the physics and chemistry at the single-atom scale, where surface characterization techniques can be applied straightforwardly. Recently there have been emerging interests in engineering materials through structural deformation or transformation. The strain field offers crucial information of lattice distortion and phase transformation in the native state or under external perturbation. Example problems with significance in science and engineering include the role of defects and dislocations in modulating material behaviors, and the process of fracture, where remarkable strain is built up in a local region, leading to the breakdown of materials. Strain is well defined in the continuum limit to measure the deformation, which can be alternatively calculated from the arrangement of atoms in discrete lattices through methods such as geometrical phase analysis from transmission electron imaging, bond distortion or virial stress from atomic structures obtained from molecular simulations. In this paper, we assess the accuracy of these methods in quantifying the strain field in 2D crystals through a number of examples, with a focus on their localized features at material imperfections. The sources of errors are discussed, providing a reference for reliable strain mapping.

## 1. Introduction

To understand mechanical processes such as structural distortion, phase transformation, fracture in two-dimensional (2D) crystals, and explore their strain-engineering applications, characterization of inhomogeneous strain distribution is crucial. For example, lattice distortion around imperfections such as point defects and dislocations in graphene were measured, which can be used to validate the continuum theory of elasticity [1,2,3]. The process of fracture in 2D crystals were analyzed through the strain distribution at crack fronts, where bond breaking and healing events are visualized [4,5]. Structures and dynamics of strained inclusion phases were characterized and discussed, revealing the microscopic mechanisms of phase transformation [6]. On the other hand, strain and phase engineering has been widely envisaged as new approaches to expand the performance spectra of 2D materials [7,8]. For example, strain gradient in graphene generates out-of-plane pseudo magnetic fields due to negligible Berry curvatures around the gapped bands at the K and K${}^{\prime }$ points [9,10]. In the continuum field theory, strain at a material point can be defined and calculated from the dimensional change of volume elements. This approach applies to the lattice representation of crystals as the Cauchy–Born approximation is valid [11]. However, quantifying inhomogeneous strain at imperfections are challenging due to the loss of lattice symmetry, as well as its localized and irregular nature.

Experimentally, atomic arrangement in 2D crystals can be visualized by transmission electron microscopy (TEM) through the contrast in imaging. The geometrical phase analysis (GPA) is a technique developed to analyze the distribution of lattice strain and rotation through the Fourier transformation of the contrast map [12], which was widely applied in the studies of quantum dots [13], nanowires [14], interfaces [15], defects [16], micro-cracks [17] and dislocations [18,19,20,21]. The idea is to identify the image-contrast maxima as lattice sites, and to calculate the local deviation from a reference lattice. The GPA relies on the assumption that the image-intensity peaks directly correspond to the positions of atomic columns in a given projection, and is especially suitable for 2D crystals with a good quality of imaging. The basic assumption of GPA is that the tangent planes to the phase images correctly approximate the geometrical phases over a small region, which is valid for slowly-varying phases [22]. The limitations of GPA beyond the basic assumption were reviewed in the literature. Error in the strain characterization arises while processing compounds [23], amorphous regions or discontinuities at material interfaces [22], lattice with a strain gradient [24] and structures with modified lattice constants from those in the available references (by substrate coupling, alloying, phase transformation, for example) [25,26]. The accuracy of GPA depends on the choice of **g**-vectors and the process of Fourier filtering (e.g., the mask size) [22], the size of reference area [27], the defocus value and the diffracted beams that determine the fringe contrast [24,28].

In theoretical studies through molecular simulations, atomic positions in 2D crystals are known by construction or evolution following the equations of motion. Strain analysis can be proceeded by measuring the distortion of inter-atomic bonds by numerical fitting an affine-transformation matrix **J** from the changes in relative positions of neighboring atoms. This approach requires the coordination of lattice sites remains unchanged during the deformation. Alternatively, strain can also be calculated from the atomistic stress with known elastic constants in the linear-response regime. This approach relies on the definition of viral stress, which remains vague [29], as well as the constitutive relations that map the strain from stress. Moreover, beyond the linear elastic limit, nonlinearity and anisotropy developed at large strain should be included in the constitutive relations [30].

The current work is focused on the validity and accuracy of the methods of strain characterization in 2D crystals. Following a brief introduction of the methodological details, strain distribution in a honeycomb lattice (i.e., the lattice of graphene, hexagonal boron nitride and transition metal dichalcogenides) with a few types of material imperfections are considered, including holes, cracks and dislocations. The strain fields obtained by GPA, bond distortion and derivation from virial stress are analyzed and compared to those predicted from continuum mechanics. We find that for some of the problems with strong strain localization, the implementation of current methods in strain analysis should be well justified before application.

## 2. Methods

### 2.1. Molecular Simulations

To obtain the equilibrium and distorted structures of graphene as an example of 2D crystals with the honeycomb lattice, molecular dynamics (MD) simulations were performed using the large-scale atomic/molecular massively parallel simulator (LAMMPS) [31]. The second-generation reactive empirical bond order (AIREBO) potential is used to describe the inter-atomic interaction [32]. Structural relaxation is carried out using the conjugate gradient (CG) algorithm with energy and force tolerance of $10^{-4}$ eV and $10^{-6}$ eV/Å, respectively. To simulate uni-axial tensile tests, the simulation box is deformed in the tensile direction, while other dimensions are free to change. Positions of atoms within the box are changed under an affine transformation, followed by structural relaxation using CG.

### 2.2. Geometrical Phase Analysis

GPA is a technique that extracts displacement information from high-resolution TEM (HRTEM) images (Figure 1a). The image intensity *I* obtained at location r for a lattice can be expanded through the Fourier series expressed in the reciprocal lattice vector g as (Bragg reflections)
(1)$$I\left(r\right)=\sum_{g}H_{g}\exp\left(2\pi ig\cdot r\right)=\sum_{g}A_{g}\exp\left\{2\pi ig\cdot r+iP_{g}\right\},$$
where $A_{g}$ and $P_{g}$ are the amplitude and phase of the coefficient $H_{g}$ at g. Inverting the FFT produces a raw geometrical phase $P_{g}^{\prime }\left(r\right)$. A reduced geometrical phase $P_{g}\left(r\right)$ can be introduced for the advantage of being rather smooth, and presenting no or only a few discontinuities, that is
(2)$$P_{g}\left(r\right)=P_{g}^{\prime }\left(r\right)-2\pi g\cdot r,$$
where the g-vector is refined to an area of homogeneous strain (e.g., the reference lattice in absence of strain). Lattice distortion, or changes in the reciprocal lattice vector g→g+Δg, will produce a gradient in the phase
(3)$$\nabla P_{g}=2\pi \Delta g.$$

The gradient in the geometric phase map $P_{g}\left(r\right)$ yields the local deviation Δg. The reduced geometrical phase map can be used to calculated the displacement field u in a crystal. For a lattice under deformation mapping r→r+u, we have
(4)$$P_{g}\left(r\right)=2\pi g\cdot u.$$

The displacement field u can be obtained from the phase map by choosing two non-collinear g-vectors ($g_{1}$, $g_{2}$) that are refined by referring to the same unstrained region through Equations (Equation 2) and (Equation 3), that is
(5)$$\left(\begin{array}{c} u_{x}\\ u_{y} \end{array}\right)=\frac{1}{2\pi }\left(\begin{array}{cc} a_{1x} & a_{2x}\\ a_{1y} & a_{2y} \end{array}\right)\left(\begin{array}{c} P_{g1}\\ P_{g2} \end{array}\right),$$
where $a_{1}$ and $a_{2}$ are the real-space basis, corresponding to the reciprocal lattice defined by $g_{1}$ and $g_{2}$, or mathematically, $a_{i}^{T}\cdot g_{j}=\delta_{ij}$. The Eulerian distortion tensor $e_{E}$ can be evaluated by numerical differentiating the displacement field u [33]. In the small-displacement regime, simplification into the linear part yields
(6)$$e_{E}=\left(\begin{array}{cc} e_{xx} & e_{xy}\\ e_{xy} & e_{yy} \end{array}\right)=\left(\begin{array}{cc} \frac{\partial u_{x}}{\partial x} & \frac{\partial u_{x}}{\partial y}\\ \frac{\partial u_{y}}{\partial x} & \frac{\partial u_{y}}{\partial y} \end{array}\right).$$

The Lagrangian representation can be obtained from the relation $I+e_{L}={\left(I-e_{E}\right)}^{-1}$, and their linearized measures are equal [22]. The strain and rotation tensors are finally obtained as
(7)$$\begin{array}{cc} \varepsilon & =\frac{1}{2}\left\{e+e^{T}\right\},\\ \omega & =\frac{1}{2}\left\{e-e^{T}\right\}. \end{array}$$

This definition can be extended to the finite-displacement regime [22]. In practice of GPA, masks are used to select Fourier components to produce phase mapping from Equation (Equation 1). In order to reduce the noise, a Gaussian mask $\exp\left[-{\left(k-g_{i}\right)}^{2}/2\sigma^{2}\right]$ can be placed around $g_{i}\left(i=1,2\;\mathrm{in}\;\mathrm{Equation}\;\left(5\right)\right)$ in the reciprocal space spanned by k. The radius of the Gaussian mask is ∼3σ, corresponding to 3/(2πσ) in the direct space. In this paper, representative radius of small (g/4) and large (g/2) values are used in GPA.

GPA in this work is performed using the Strain++ code [12,23,34]. The images used in GPA are generated from atomic structures, where atoms are represented in balls with a radius of 0.06 nm [35], and converted into the 8-bit tagged image file format (TIFF) using ImageJ [36].

### 2.3. The Bond Method

For the discrete atomic presentation of 2D crystals, the strain field measures deformation from the difference between reference (initial, undeformed) and current structures (Figure 1b). Specifically, the relative position between the atom *i* and its neighbor *j* are calculated as $d_{ji}=x_{j}^{0}-x_{i}^{0}$ and $d_{ji}=x_{j}-x_{i}$ at the reference and current configurations, respectively. The affine-transformation matrix J is fitted for the best mapping $\left\{d_{ji}^{0}\right\}\to \left\{d_{ji}\right\}$ [37,38], that is
(8)$$J_{i}={\left(\sum_{j\in N_{i}}d_{ji}^{0T}d_{ji}^{0}\right)}^{-1}\left(\sum_{j\in N_{i}}d_{ji}^{0T}d_{ji}\right),$$
where $N_{i}$ is the number of neighbors of atom *i* ($N_{i}=3$ for the honeycomb lattice). As an example, for graphene we have $\sum_{j\in N_{i}=3}d_{ji}^{0T}d_{ji}^{0}=1.5a_{C-C}^{2}I$, where $a_{C-C}$ is the C-C bond length in graphene, then $J_{i}$ is written as
(9)$$J_{i}=\frac{1}{1.5a_{C-C}^{2}}\sum_{j\in N_{i}}d_{ji}^{0T}d_{ji}.$$

This method also works for problems with no reference structures such as a lattice with dislocations embedded, as the neighbors are identified and remains unchanged during deformation. The Lagrangian strain tensor is finally calculated for the atom *i* as
(10)$$\varepsilon_{i}=\frac{1}{2}\left(J_{i}J_{i}^{T}-I\right).$$

### 2.4. The Stress Method

The atomistic stress can be calculated following the virial definition [29], from the instantaneous atomic positions $\left\{r_{i}\right\}=r_{1},...,r_{N}$ and their interaction (Figure 1c). For systems with two-body interaction, the virial contribution is
(11)$$W_{ab}=\frac{1}{2}\sum_{n=1}^{N_{p}}\left(r_{1a}F_{1b}+r_{2a}F_{2b}\right),\mathrm{where}\;a,b=\left(x,y,z\right),$$
where $N_{p}$ is the number of interacting pairs. $F_{1}$ and $F_{2}$ are the forces exerted on atoms $r_{1}$ and $r_{2}$ in the pair. For many-body potentials, the virial is summed over the atoms in groups (bond angles, dihedrals, etc.), and then divided evenly among those atoms [29]. The atomistic stress is finally obtained by dividing the virial by the per-atom volume.

The small-displacement strain field is obtained through stress–strain relations under the plane-stress assumption as
(12)$$\left(\begin{array}{c} \varepsilon_{xx}\\ \varepsilon_{yy}\\ \varepsilon_{xy} \end{array}\right)=\frac{1}{E}\left(\begin{array}{ccc} 1 & -\nu & 0\\ -\nu & 1 & 0\\ 0 & 0 & 1+\nu \end{array}\right)\left(\begin{array}{c} \sigma_{xx}\\ \sigma_{yy}\\ \sigma_{xy} \end{array}\right),$$
where a linear, isotropic, elastic constitutive model is used, and graphene is considered as a plate with thickness of 0.34 nm. The Young’s modulus and Poisson’s ratio of graphene are 960 GPa and 0.35, which are obtained from our MD simulations using the AIREBO potential. For nonlinear and anisotropic elasticity that are activated at uni-axial strain of ∼5% ad ∼15%, respectively, a constitutive model with higher-order elastic constants is required [39,40].

## 3. Results

### 3.1. Uniform Strain and Uniform Strain Gradient

We first validate the methods of strain characterization for samples with uniform strain or a uniform strain gradient, by applying uni-axial stretch to rectangular and trapezoidal graphene monolayers that can be shaped by focused ion beams (FIB) in experiments [41]. In GPA calculations, a deformed structure of graphene is patched to a fully-relaxed one, which is used as the reference (Figure 2a). The strain is measured using a mask with size of g/4 or g/2 [22,24]. The results show that using the smaller mask suppresses the noise, but local features such as the interface between the reference and deformed lattice is less distinct (Figure 2b). By comparing the strain values characterized at different loading amplitudes (Figure 2c), we find that GPA and bond methods yield accurate measure. The difference between Euler and Lagrange strain measured in GPA is noticeable, especially under large deformation. In contrast, the strain derived from virial stress is accurate only under small stretch for the linear and isotropic assumption in material responses. Under large deformation, the deviation is significance, signaling that the nonlinear, anisotropic constitutive model have to be implemented in the method [30].

For the stretched trapezoidal sample, strain distribution is analyzed along the axis of mirror symmetry (Figure 2d–f). The tensile strain is expected to increase linearly with decreasing cross-section area. The strain gradient is a constant across the sample, except for the localization at the clamping ends due to the effects of boundary constraint. Our results show that GPA yields almost identical strain fields with that obtained from the bond or stress method in the central region of the sample. Deviation in the strain values is identified at the loading boundaries, where the maximum strain occurs at different locations (Figure 2e,f). The strain field mapped out of the sample region demonstrates the deficiency of GPA in analyzing non-periodic data. As a result, the boundary of region occupied by the samples has to be contoured manually. These results (uniform strain or uniform strain gradient) demonstrate the validity of the strain-characterization methods for 2D crystals with relatively smooth strain distribution. We will then extend the study to structures with localized distortion.

### 3.2. Strain Concentration at Holes

Circular holes is one of the most common defects considered in the discussion of strain or stress concentration in structures under mechanical loads. We apply uni-axial stretch (along *x* with a nominal strain of ε) to a graphene monolayer of 50 nm × 50 nm (Figure 3a). A single circular hole with a radius of a=2 nm is created in the center of sample. Periodic boundary conditions (PBCs) are applied along the *x* and *y* directions in the MD simulations. The strain component $\varepsilon_{xx}$ around the hole at ε=3.7% and tensile stress of σ=34 GPa is characterized by using the GPA, bond and stress methods (Figure 3c–f). The results are compared to the prediction from the theory of linear elasticity, which are, in polar coordinates (r,θ) [42]
(13)$$\begin{array}{cc} \sigma_{r} & =\frac{1}{2}\sigma \left(1-\frac{a^{2}}{r^{2}}\right)+\frac{1}{2}\sigma \left(1-\frac{4a^{2}}{r^{2}}+\frac{3a^{4}}{r^{4}}\right)\cos\left(2\theta \right),\\ \sigma_{\theta } & =\frac{1}{2}\sigma \left(1+\frac{a^{2}}{r^{2}}\right)-\frac{1}{2}\sigma \left(1+\frac{3a^{4}}{r^{4}}\right)\cos\left(2\theta \right),\\ \tau_{r\theta } & =-\frac{1}{2}\sigma \left(1+\frac{2a^{2}}{r^{2}}-\frac{3a^{4}}{r^{4}}\right)\mathrm{sim}\left(2\theta \right). \end{array}$$

The distribution of $\varepsilon_{xx}$, obtained from coordinate transformation, shows similar contours of elongation and compress regions for all the methods. In detail, the $\varepsilon_{xx}$ plotted along the 1−1 cross section (Figure 3b) increases rapidly while approaching the hole edge. The value of $\varepsilon_{xx}$ calculated from the bond or stress method is 3.7% in the region far from the hole edge, which is slightly underestimated as 3.4% and 3%±0.3% by GPA using the mask with a size of g/4 and g/2, respectively. At the hole edge, significant deviation from the theoretical prediction is identified for all of the three methods. The strain concentration factor $\varepsilon_{\max}/\varepsilon$ is measured to be 3, 3.25, 3.76 and 2.74 from the theory of linear elasticity, GPA, bond and stress methods, respectively. The GPA result is sensitive to the size of mask. The overestimation of strain from the bond method is due to the presence of open edges, where strain is localized to several atoms at the edges and that within the sp${}^{2}$ lattice is much smaller. Besides of assumptions in the linearized constitutive model, strain underestimation in the stress method may also comes from the definition and calculation of virial stress at the edge. It is also interesting to note that, by comparing the strain calculated from the bond and stress methods, the atoms bearing the largest stress is not the ones with largest distortion, which may be attributed to the discrete and incomplete nature of lattice at the hole edges.

### 3.3. Strain Fields at the Crack Tip

Knowledge of the strain and stress fields at crack-tips is critical to understand the fracture of solids, a process of multiscale nature from bond breaking at the atomic scale to the far field of strain and stress at the continuum level [43]. One advantage of 2D crystals in the study of fracture mechanics is that all of their constituting atoms are are exposure to the environment, and can be directly characterized by various forms of probing in experiments. To characterize the strain field, we introduce a pre-crack in the graphene sample in the middle of left edge, which has a width of one lattice constant and aligns along zigzag direction. In the MD simulations, the PBC is enforced in the direction (*y*) in perpendicular to the crack, along which the displacement load is applied by deforming the simulation box (Figure 4a). Other boundaries are open. In linear elastic fracture mechanics (LEFM), the solution of displacement field near crack-tip for this Mode-I setup is
(14)$$\begin{array}{cc} u_{x} & =\frac{K_{I}}{2\mu }\sqrt{\frac{r}{2\pi }}\cos\left(\frac{\theta }{2}\right)\left[\kappa -1+2\sin^{2}\left(\frac{\theta }{2}\right)\right],\\ u_{y} & =\frac{K_{I}}{2\mu }\sqrt{\frac{r}{2\pi }}\sin\left(\frac{\theta }{2}\right)\left[\kappa +1-2\cos^{2}\left(\frac{\theta }{2}\right)\right], \end{array}$$
where $\kappa =\frac{3-\nu }{1-\nu }$ for plane-stress problems, and r=(r,θ) is the position measured from the tip. The stress intensify factor (SIF) is $K_{I}=\sigma_{y}\sqrt{\pi a}=4.38\;\mathrm{MPa}\sqrt{m}$, where a=6.6 nm and $\sigma_{y}=30.4$ GPA are obtained from the model setup and MD simulations under applied nominal strain of ε=3.4% [44]. The contours of strain component $\varepsilon_{yy}$ calculated using the bond and stress methods show the two-lobe features, in consistency with the LEFM prediction. GPA, however, shows a rounded contour around the tip (Figure 4c–f), signaling the failure in extracting the crack-tip strain fields. Moreover, the issue of open boundaries in GPA is also identified through the unphysical results in the crack region and at the edges of samples. Plotting $\varepsilon_{yy}$ from the crack tip and along the *x*-direction (y=0) (Figure 4b) shows that, the far-field values of $\varepsilon_{yy}$ is 3.4% by using the LEFM theory, and the bond, stress methods, aligning well with the loading condition. GPA with g/4 and g/2 estimates 3% and 3.5%, respectively, showing the dependence on the mask size.

The strain fields at the crack tip calculated using different methods capture the divergent nature as predicted in LEFM. The crack surfaces are defined by the positions of edge atoms (plus a boundary layer with thickness of $\frac{1}{2}a_{C-C}$) by considering the discrete nature of lattices. The maximum relative bond elongation in the structure is 40%. In our calculations, GPA predicts peak strain of 22% and 20% for g/4 and g/2, while the bond and stress methods estimates 24.4% (average of 37.4% and 11.4% in the lattice at crack-tip) and 9.4% (average over 10.8% and 8%), respectively. The difference in strain variation at the crack tip clearly highlights the loss of accuracy in characterizing strain in atomic structures with highly localized and inhomogeneous strain fields. Alternatively, the bond method offers a direct measure of structural deformation in the discrete representation, which could be a better choice for the use in the study of fracture processes [45].

### 3.4. Lattice Distortion by Dislocations

Dislocations break the symmetry of translational invariance in crystals, resulting in localized lattice distortion, as well as stress and strain buildup within the material. The distortion induced by pentagon-heptagon (5|7) pairs can be considered as the core of in-plane edge dislocation in the honeycomb lattice. Their arrangement in arrays is then a grain boundary (GB). Previous studies show that the stress distribution around these topological defects can be reasonably predicted by the linear elastic theory of dislocations except for that in the core region [3,46]. It should also be noted that, the out-of-plane distortion of 2D crystals could further release the in-plane stress or strain. In this work, both the isolated dislocation and an array of them are analyzed using the strain-characterization methods. For isolated dislocations, two dislocation cores with opposite Burgers vector are placed 12 nm away from each other. PBCs are enforced in the in-plane dimensions, but only half of the sample with a single dislocation is used for strain analysis (Figure 5a). The strain field extracted using GPA with a mask size of g/2 and the stress method are compared with theoretical predictions [3]. For GPA calculations with as small twist angle ϕ across the GB (Figure 5b), the g-vectors of the two domains are close, and the strain field can be well characterized. Theoretically, the displacement of edge dislocation given by linear, isotropic elastic theory [20] is
(15)$$\begin{array}{cc} u_{x} & =\frac{b}{2\pi }\left[\tan^{-1}\frac{y}{x}+\frac{xy}{2\left(1-\nu \right)\left(x^{2}+y^{2}\right)}\right],\\ u_{y} & =-\frac{b}{2\pi }\left[\frac{1-2\nu }{4(1-\nu )}\ln\left(x^{2}+y^{2}\right)+\frac{x^{2}-y^{2}}{4\left(1-\nu \right)\left(x^{2}+y^{2}\right)}\right], \end{array}$$
and the displacement given by the Foreman model [20,47] is
(16)$$u_{x}=-\frac{b}{2\pi }\left[\tan^{-1}\frac{2(1-\nu )x}{ay}+\left(a-1\right)\frac{2(1-\nu )xy}{4{\left(1-\nu \right)}^{2}x^{2}+a^{2}y^{2})}\right],$$
where $b=\sqrt{3}a_{C-C}$ is the length of the Burgers vector of graphene. *a* is a fitting parameter in the Foreman model, which is reduced to the Peierls–Nabarro (PN) model with a=1 [20,48]. From the strain component $\varepsilon_{xx}=\frac{\partial u_{x}}{\partial x}$ measured using GPA, bond, stress and theories (Figure 5b–f), we find that the distribution from GPA exhibits to be nearly round, and in good accordance with the Foreman model by choosing a=3, failing to capture the butterfly feature shown in the results from the bond and stress methods. Moreover, from the distribution of $\varepsilon_{xx}$ along the 1−1 cross-section, all methods are able to capture the slop of strain gradient near the dislocation. However, in GPA and theoretical predictions, strain at the centers of dislocations shows singularity, signaling the nature of continuum representations. In the bond and stress methods with discrete atomic structures, the maximum and minimum strain values are finite, which are calculated to be 22%, −9% and 11%, −11%, respectively.

For GBs with significant misorientation (Figure 5g), the g-vector spots (Bragg beams) of each domain are well separated in the reciprocal space. The masks applied in GPA have to cover the peaks corresponding to the g-vectors in neighboring domains. For the honeycomb lattice, the distance between the two sets of g-vectors is $\phi \cdot g\in \left[0,0.5236g\right]$ for $\phi \in \left[0,30^{\circ }\right]$. To cover two g-vectors chosen in neighboring domains, the size of mask has to be increased by ϕ·g at least, if the mask is centered at one of the chosen g-vectors. The reduced geometrical phase $P_{g}\left(r\right)$ obtained form Equation (Equation 2) yields the lattice distortion. Masking two g-vectors in the same domain should be avoided, which is not an issue for domains with a small angle of twist. However, for significant misorientation, the mask should be centered at the middle point between the two g-vectors. A critical angle of twist can be estimated as ∼15${}^{\circ }$, for example, from ϕ·g=g/4.

For low-angle grained boundaries (LAGBs) with $\phi =5^{\circ }$, the distance between the g-vectors is small, and the strain distribution is derived from $P_{g}\left(r\right)$. The strain distributions measured from GPA and the bond, stress methods are almost identical with that obtained by overlapping strain fields around isolated dislocations (Figure 5h,i). For high-angle GBs (HAGBs) with $\phi =30^{\circ }$ as another example, the strain and rotation maps are obtained with a mask with size of g/2, centered at the middle point between two selected g-vectors (Figure 5j). The results are almost identical with those calculated from the bond and stress methods, and the patterns of strain localization align well with the locations of (5|7) pairs. In HAGBs with a high density of dislocations, interaction between them modifies the strain distribution, which can be shown by comparison with the results of isolated dislocations. GPA also predicts infinite strain localization due to its continuum nature, while bond and stress methods yield finite values.

## 4. Discussion

Digital image correlation (DIC) [49] and Raman spectroscopy [50] are also commonly-used techniques for strain characterization in materials science. In DIC, strain is measured by tracking the correlation between displacements of feature speckles on the surface of samples, such as the immobile contaminants. The accuracy is defined by the number, size of speckles, and the algorithm to analyze the correlation. Raman spectroscopy measures strain, in graphene for example, through the shift of G and 2D peaks. The spatial resolution of Raman mapping is limited by the size of laser spots (∼500 nm), and the extracted values of strain are averaged over the spot. Consequently, these methods apply only for strain characterization at the micrometer scale. To resolve the mechanical responses of materials at imperfections, the spatial resolution of strain characterization has to be elevated.

As the resolution of experimental imaging approaches the atomic scale, strain measured from the discrete sites of atoms offers the alternative measure of structural distortion. Our results show that deformation with uniform strain or a uniform strain gradient can be quantitatively well captured by the methods of GPA, bond and stress. However, localized, inhomogeneous deformation at material imperfections not only signals the failure of linear-elastic theory [51,52], but also pose challenges to strain characterization.

The GPA method reports continuous strain field from the FFT map of lattices. The intrinsic assumption made in GPA requires slowly-varying geometrical phases. At the very first image-filtering step, the size and shape of masks matters the most. A Gaussian mask with a radius in the range between g/4 and g/2 is used in this work for comparative studies. To measure strain in regions with a small gradient, a smaller mask size is more accurate, while for structures with strain localization, a larger mask size should be used to capture stronger strain variation. To capture the deformation between domains with significant misorientation, the size and position of masks should be specified to include the g-vectors in each domain. GPA also suffers from the difficulty in determining the geometrical boundaries for finite-size samples, which should be manually determined if the strain value at the boundary is needed.

With the reference configuration of lattices known for 2D crystals, the bond or stress method reports strain measures of the lattice. In the bond method, rotation is excluded, and it required the coordination of lattice sites should remain unchanged during deformation, which depends on the definition of critical bond lengths. The stress method relies on the definition of virial stress and constitutive models, which remain challenging for many-body interactions and nonlinear, anisotropic responses at large strain [40]. The assignment of many-body contributions into participant atoms remains vague, which is another source of errors [53]. For lattice distortion that is difficult to be determined by GPA at high accuracy, one could extract the atomic positions from the high-resolution images of experimental characterization, and then apply the bond method or the stress method if MD simulations are performed for the resolved structures with reliable interatomic potentials. A critical issue to be addressed in this approach is to reduce the impact of noise and faults in the imaging process, which could be improved by, for example, the machine learning techniques [54,55].

All of the three methods assessed in this work exclude the out-of-plane deformation of 2D crystals, which could be significant once the topological defects are introduced [56]. GPA is based on 2D mapping of the lattices and cannot accommodate information in the extra dimension. Bond and stress methods can partly include the effect since the bond distortion and atomistic stress can be modified by the out-of-plane deformation. Adequate measures should take into account for the local curvature and bending deformation, in addition to the in-plane strain components.

## 5. Conclusions

In summary, we assessed the strain-characterization methods in analyzing the strain distribution in 2D crystals, including GPA, bond-distortion analysis, and derivation from the virial stress. Example problems including strain localization at holes, crack tip, dislocations and grain boundaries are analyzed. Our discussion on the sources of errors offers guidance for strain characterization for 2D crystals, which play an important role in elucidating mechanical responses of materials at nanoscale. These strain measures were also extended recently to the interfaces between 2D crystals, as well as their multilayers and heterostructures [57,58]. The combination of the above-mentioned methods thus offers a flexible and powerful approach to understand their mechanical behaviors and design strain- or phase-engineering applications.

## Figures and Tables

**Figure 1 materials-14-04460-f001:**
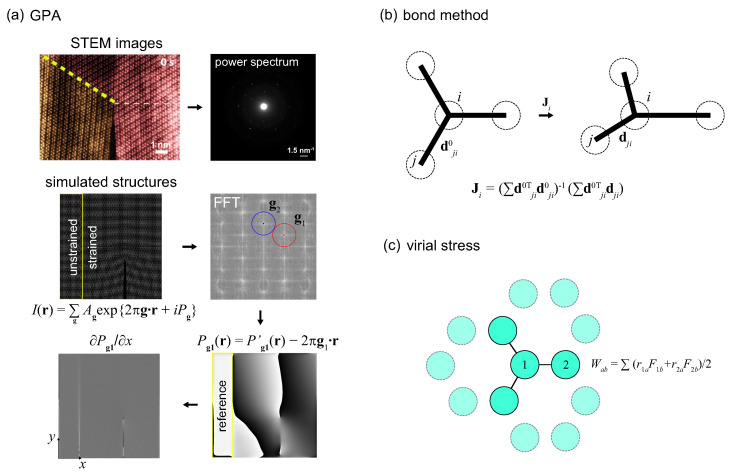
Methods of strain characterization based on (**a**) GPA, (**b**) bond distortion and (**c**) virial stress. The scanning TEM (STEM) image in panel (**a**) is adapted from [4] under the Creative Commons Attribution 4.0 International license.

**Figure 2 materials-14-04460-f002:**
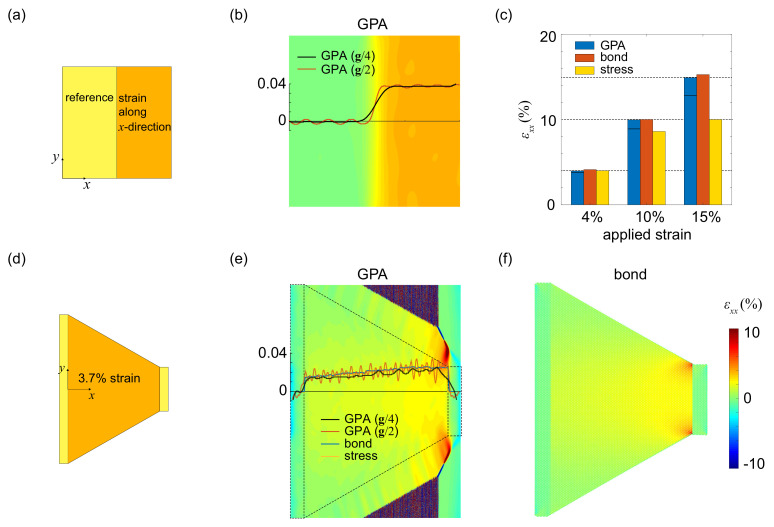
(**a**) A composite structure by patching an undeformed lattice (**left**) to a uniformly stretched one (**right**, strain along the *x* direction). (**b**) Uni-axial strain field calculated from GPA at 4% strain. Strain distribution along *x* is plotted as the inset in panel (**b**) for GPA using mask size of g/4 or g/2. (**c**) Lagrangian strain estimation from GPA, bond and stress methods at strain of 4%,10%,15%. The Eulerian strain from GPA are plotted the solid lines in the bar representation. (**d**) Strain characterization for a stretched trapezoidal sample with a uniform strain gradient. Left and right ends (2 nm width) of the graphene lattice are displaced to apply a nominal strain of ε=3.7% along the *x* direction. (**e**,**f**) Uni-axial strain fields calculated from (**e**) GPA and (**f**) bond methods. Strain distribution along the axis of mirror symmetry is plotted as inset in panel (**e**) for all the three methods.

**Figure 3 materials-14-04460-f003:**
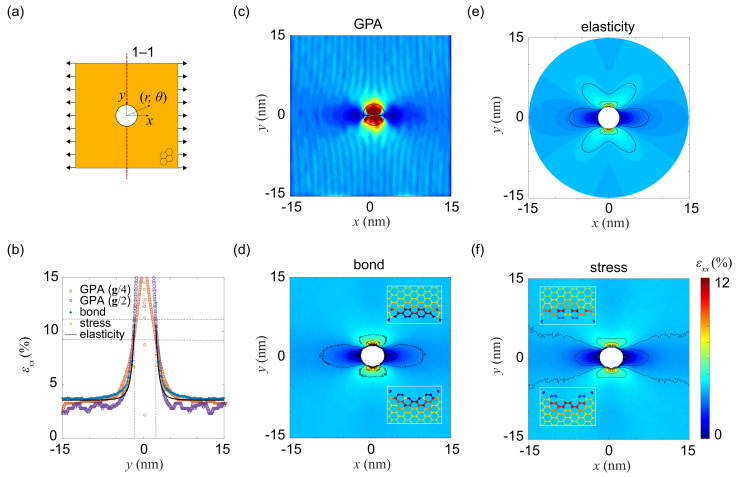
Strain characterization for a graphene monolayer with a circular hole. (**a**) Uni-axial strain is applied along the *x*-direction. (**b**) Distribution of strain component $\varepsilon_{xx}$ along the 1−1 cross section annotated in panel (**a**). (**c**–**f**) $\varepsilon_{xx}$ map calculated from (**c**) GPA, (**d**) bond distortion, (**e**) the theory of linear elasticity, and (**f**) virial stress.

**Figure 4 materials-14-04460-f004:**
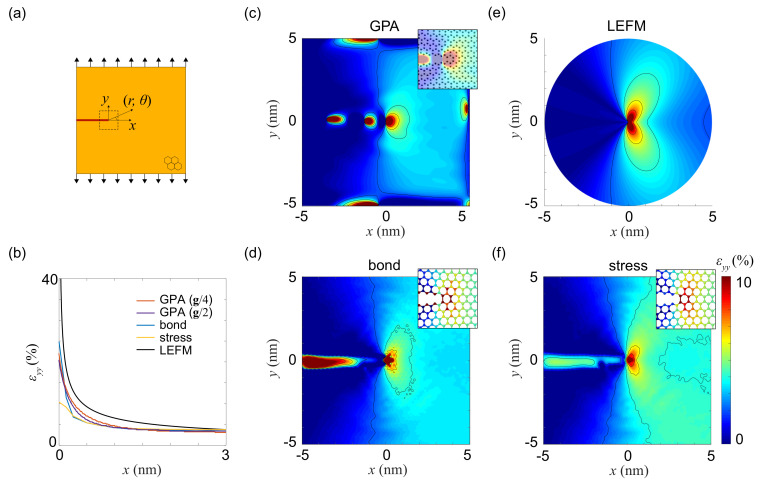
Strain characterization for a mode-I crack tip. (**a**) A pre-crack is created at the middle of left edge in the graphene monolayer. Strain is applied along *y* (the armchair direction) by deforming the simulation box. (**b**) $\varepsilon_{yy}$ plotted along *x* from the tip. (**c**–**f**) $\varepsilon_{yy}$ map calculated from (**c**) GPA, (**d**) bond distortion, (**e**) LEFM, and (**f**) virial stress.

**Figure 5 materials-14-04460-f005:**
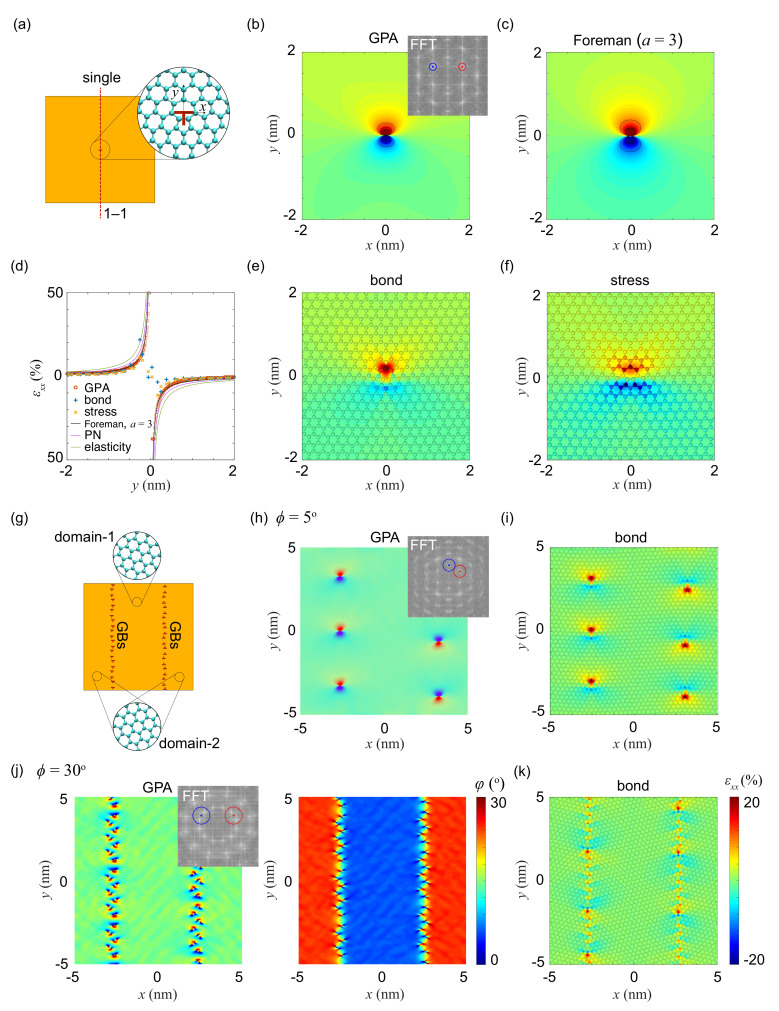
Strain characterization for isolated dislocations (**a**–**f**) and GBs (**g**–**k**). (**a**–**f**) $\varepsilon_{xx}$ map calculated from (**b**) GPA, (**c**) the Foreman model with a=3, (**e**) bond distortion, (**f**) virial stress. Panel (**d**) summaries $\varepsilon_{xx}$ along the cross section 1−1 annotated in panel (**a**), which also includes the theoretical predictions from isotropic linear elasticity and the Peierls–Nabarro model. (**g**–**k**) Strain component $\varepsilon_{xx}$ and in-plane rotation φ in panel (**j**) map calculated from (**h**,**j**) GPA and (**i**,**k**) bond distortion. Results are plotted for GBs with the twist angle between the neighboring domains of $\phi =5^{\circ }$ (**h**,**i**) and $\phi =30^{\circ }$ (**j**,**k**).

## Data Availability

The data presented in this study are available on request from the corresponding author. The data are not publicly available as raw/processed data required to reproduce these findings because the data cannot be shared at this time as it also forms part of an ongoing study.

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
