# Peer review of "Strain Characterization in Two-Dimensional Crystals"

_materials, 2021, doi:10.3390/ma14164460_

Round 1
Reviewer 1 Report
Methods for quantifying strain components in 2D crystal
lattices are compared. The experimental GPA method uses
displacement data from HRTEM followed by differentiation.
The bond method uses relative positions of atoms between
deformed and initial configurations to compute the Lagrangian
strain tensor. The stress method computes the virial stress
from atomic simulation, then inverts the constitutive law
to solve for strain. The latter two methods apply to
simulation data at the atomic scale. In this paper,
differences among the methods are compared for graphene.
Cases consider uniform strain, uniform strain gradient,
a circular hole, a mode I crack tip, and an edge dislocation.
The simulation results seem to be in general agreement
with TEM, but some differences arise, as expected, near
sources of strain concentration or singularities.
This paper provides a nice summary of three methods
of strain characterization. The data and numerical
results seem new, and insight into the limitations of
the methods is a useful contribution to the community.
The figures are of high quality.
Publication is recommended after the following rather
minor points can be addressed:
[1] The GPA method, which uses TEM data, seems to be
restricted to the nanoscale. It should be mentioned if
there are any other valid
methods for strain characterization, for example
Digital Image Correlation (DIC), that might be used
at larger length scales of resolution.
[2] Isotropic linear elasticity is used to model
graphene in eq. (12). It would be beneficial for
the authors to list the full set of anisotropic
elastic constants for the material (for example,
from literature if available) so that the
accuracy of the isotropic assumption can be
quantified.
[3] The discussion about finite strain effects
seems inconsistent, and perhaps not fully rigorous.
In the regime of small displacements and small
displacement gradients, the Eulerian and Lagrangian
tensors are equal, and eqs. (6) and (7) are correct.
However, the tensor in equation (6) is the
linear strain tensor,
and it is not Eulerian finite strain tensor of Almansi,
Finger, or Murnaghan. This should be clarified
in the manuscript. The paper can also recommend some textbook
references on the geometrically nonlinear theory, for example:
-A.C. Eringen, Nonlinear Theory of Continuous Media,
McGraw-Hill, NY (1962)
-J.D. Clayton, Nonlinear Mechanics of Crystals,
Springer, Dordrecht (2011)
[4] It should be noted that the linear elastic solutions
for strain fields in the vicinity of singularities such
as crack tips and dislocation cores are probably
inaccurate very close to the defect
since the elastic fields diverge at such singularities
as theoretical continuum strains become large.
Some references on study of dislocations and defects in
nonlinear elastic crystals with analytical solutions
might be added in this regard.
[5] Figure 5 caption, Foreman is missing the "n".
Reviewer 2 Report
Authors report strain-characterization methods in analyzing the strain distribution in 2D crystals, including GPA, bond-distortion analysis, and derivation from the virial stress. The paper is very well written and can be published in Materials.
Reviewer 3 Report
The work entitled "Strain Characterization in Two-Dimensional Crystals" is a very interesting work that describes the accuracy of the quantification from the strain field in 2D crystals. The analysis of the strain localization at holes, crack tip, dislocations, and grain boundaries can open new directions on such types of materials, and understand their mechanical properties. I will suggest the acceptance of the article after some minor English spelling.
